REGISTERED REPORT PROTOCOL

# Comparing the influence of stimulus size and contrast on the perception of moving gratings and random dot patterns—A registered report protocol

Benedict Wild[1,2]*, Stefan Treue[1,3,4,5]

**1** Cognitive Neuroscience Laboratory, German Primate Center – Leibniz-Institute for Primate Research, Goettingen, Germany, **2** Goettingen Graduate Center for Neurosciences, Biophysics, and Molecular Biosciences (GGNB), University of Goettingen, Goettingen, Germany, **3** Faculty of Biology and Psychology, University of Goettingen, Goettingen, Germany, **4** Leibniz-ScienceCampus Primate Cognition, Goettingen, Germany, **5** Bernstein Center for Computational Neuroscience, Goettingen, Germany

* bwild@dpz.eu

This is a Registered Report and may have an associated publication; please check the article page on the journal site for any related articles.

## Abstract

Modern accounts of visual motion processing in the primate brain emphasize a hierarchy of different regions within the dorsal visual pathway, especially primary visual cortex (V1) and the middle temporal area (MT). However, recent studies have called the idea of a processing pipeline with fixed contributions to motion perception from each area into doubt. Instead, the role that each area plays appears to depend on properties of the stimulus as well as perceptual history. We propose to test this hypothesis in human subjects by comparing motion perception of two commonly used stimulus types: drifting sinusoidal gratings (DSGs) and random dot patterns (RDPs). To avoid potential biases in our approach we are pre-registering our study. We will compare the effects of size and contrast levels on the perception of the direction of motion for DSGs and RDPs. In addition, based on intriguing results in a pilot study, we will also explore the effects of a post-stimulus mask. Our approach will offer valuable insights into how motion is processed by the visual system and guide further behavioral and neurophysiological research.

## Introduction

Our visual world is highly dynamic and the ability to perceive motion is essential for surviving and striving in an ever-changing world. In addition to its everyday relevance, motion processing is also an excellent and well-studied model system for visual neuroscience more generally. Linear motion can be characterized by its direction and speed and a large number of studies have attempted to describe how these parameters are linked to fundamental neuronal properties [1–3]. Furthermore, a limited number of well-defined regions in primate visual cortex are known to respond selectively to these parameters. The often implicitly assumed "standard model" of motion processing assigns key roles in representing direction and speed to neural

**Competing interests:** The authors have declared that no competing interests exist.

activity in primary visual cortex (V1) and the middle temporal (MT) area [4]. Visual area 3 (V3) [5] and the medial superior temporal cortex (MST) [6–9] are further regions that play important roles for the processing of motion information. Direction-selective cells in V1 have small receptive fields and can therefore only provide information about local aspects of motion. In consequence, these cells suffer from the "aperture problem": they can only detect motion in a direction that is orthogonal to any edge or border that extends beyond their receptive fields [10–13]. These direction-selective V1 cells provide the main input to area MT [14] where cells have larger receptive fields, integrate over larger regions of the visual field, and can therefore solve the ambiguities that is inherent in the activity of V1 neurons. The importance of area MT has been highlighted by numerous studies, suggesting that it is both necessary and sufficient for motion perception. Lesions in area MT lead to increased motion thresholds [15, 16] while micro-stimulating neurons in this area can bias a monkey's judgments of motion direction [17, 18], suggesting a direct link between MT activity and motion perception. Based on these results it is broadly agreed that in the healthy brain motion perception relies on signals that are processed first by V1 and then, subsequently, by MT (e.g., [11, 13, 19, 20]. This "standard model" has recently been questioned, however, suggesting that alternate perceptual pathways that bypass area MT can be accessed under special circumstances [21]. In this study monkeys were trained over several weeks to indicate the direction of a briefly presented Gabor grating of varying contrast levels. Recordings in MT confirmed that cells in that area were tuned to the direction of the gratings and that neuronal activity was correlated to behavior. Surprisingly, however, reversible inactivation of MT had little effect on behavioral performance, indicating that MT is not always necessary for motion perception and that other motion-sensitive areas (e.g., V1) can compensate for the disruption. Monkeys were then trained for several weeks on a similar task that used random dot patterns (RDPs) of varying coherence levels rather than gratings of varying contrast levels. Again, MT activity was correlated with behavioral performance, but this time MT inactivation did affect performance. When the grating experiment was repeated after the training with RDPs, MT inactivation, which had previously had very little effect on behavior in the grating task, led to a large performance impairment. The authors concluded that prolonged training with random dot stimuli increased the contribution of area MT to perceptual decisions about motion, even for other types of motion stimuli. In addition to the effects that inactivating area MT had on motion perception, Liu & Pack [21] also report a behavioral signature for a shift of perceptual processing between different brain regions: when testing the monkeys' motion discrimination performance on high-contrast gratings of varying sizes, performance increased with stimulus size up to a certain point (the "optimal stimulus size"), before it declined for even bigger stimuli. This "spatial suppression" effect was first described by Tadin and colleagues [22] and has been repeatedly replicated in humans [23–29] and monkeys [30]. Originally it had been hypothesized that this effect is caused by the suppressive surround of receptive fields of individual MT neurons which is stimulated by gratings exceeding a certain size [22]. Physiological recordings and modeling, however, suggest that it is the surround suppression of several neurons and the correlation in activity between these neurons rather than suppression of individual neurons that causes the psychophysical suppression effect [30]. In their more recent study, Liu and Pack [21] found that the "optimal stimulus size" for gratings increased after training with RDPs, which they interpret as evidence that the prolonged training with the RDPs shifted perceptual processing to an area with larger receptive fields (e.g., from V1 to MT). Such a shift in optimal stimulus size could potentially serve as an easily obtained measure for which brain regions are primarily involved in motion perception in a given task. Together, these results suggest that there is not a single processing pipeline for visual motion that works in the same way for all types of stimuli. Instead, the contributions of different motion-sensitive brain areas

seem to be highly flexible and the particular features of a stimulus appear to play a role in determining which areas or local neuronal networks are recruited for the task at hand. This is in line with the "Reverse Hierarchy Theory" [31, 32] which suggests that neurons at different levels of the sensory processing pathways are recruited as needed to optimize perception and perceptual learning. We aim to systematically evaluate the hypothesis of a stimulus-dependent neural processing of visual motion information by establishing and comparing the effects and interactions of stimulus size and contrast on the perception of the two best established motion stimulus types: drifting sinusoidal gratings (DSGs) and random dot patterns (RDPs). If the motion of DSGs can be achieved without MT, whereas the perception of RDPs requires activity in MT, as the results by Liu & Pack [21] suggest, we hypothesize to see a larger optimal stimulus size for RDPs than for DSGs. In a pilot experiment (S1 Appendix) we were able to to replicate some, but not all of the previous findings with regard to the effect of stimulus size, and how it depends on contrast. A major difference in our experiment, compared to previous studies, was the use of a post-stimulus mask, which we had included to prevent visual persistence of the briefly presented stimuli (see Discussion in S1 Appendix). To determine whether a post-stimulus mask prevents some of the previously reported spatial suppression effects (which would provide important constraints for the interpretation of the effects) we will include this as a variable in our study.

## Materials and methods

### Participants

We will recruit participants for our study from a subject pool at the Göttingen Campus. To determine the sample size, we ran a power analysis based on results from a pilot study (S1 Appendix and S1 Dataset). As we had no data to estimate the effect of a post-stimulus mask, we simulated the response under the assumption that the mask would decrease the overall performance, but would not interact with any of the other three predictor variables (stimulus type, stimulus size, contrast). We compared a full-model that included main effects of stimulus type, stimulus size, contrast, and mask as well as their interactions up to order four with a null model that only included the effect of mask. We simulated experiments with 12, 18, 24, or 32 participants, running 100 simulations of each experiment. For all four sample sizes, the difference between the full model and the null model was significant for each simulated data set. We therefore decided to collect data from 18 participants, even though 12 participants would be sufficient to detect a significant difference between a full model and a null model, because the increased sample size will increase the precision of the estimates and allow us to detect main effects and interactions with a smaller effect size. Participants have to fulfill the following recruitment criteria: in total, they need to be gender balanced, each subject needs to be naïve as to the exact purpose of the study, be between 18 and 35 years old, and have normal or corrected to normal vision. Given the ongoing COVID-19 pandemic, we also ensure that subjects show no relevant symptoms (such as increased temperature as measured with infrared contactless thermometer, coughing, running nose, etc.). We decided to exclusively focus on young adults because previous studies found that the suppression effect in large, high-contrast gratings varies with age, with older participants showing improved performance in some tasks [23]. If individual participants' data cannot be analyzed (for example, because of technical issues during the data collection), replacement subjects will be recruited to ensure that data from 18 participants is available for data analysis. All participants will be asked to give informed written co¬nsent prior to participating in the study. The study as described here adheres to institutional guidelines for experiments with human subjects, which has been approved by the Ethics Committee of the Georg-Elias-Müller-Institute of Psychology,

University of Göttingen (GEMI 17–06-06 171), and is in accordance with the principles of the Declaration of Helsinki.

## Experimental setup

The experiments will be conducted in a dimly-lit room and stimuli will be presented on an LCD screen (SyncMaster 2233, Samsung) with a refresh rate of 120 Hz. The experiment will be controlled with the open-source software MWorks (mworks-project.org) running on an Apple MacPro computer. Subjects will be asked to respond on a gamepad (Precision, Logitech). All stimuli will be presented at the center of the screen (see the following section), and subjects will be provided with a central fixation point between trials. We will track subjects' eye movements using an Eyelink 1000 system (SR-Research, Ottawa, ON, Canada) at a sample rate of 500 Hz. Subjects' heads will be stabilized with a chin-and-head rest that is positioned 57 cm in front of the monitor.

## Stimuli and procedure

Subjects will be presented with either a horizontally drifting, sinusoidal grating with a 2D Gaussian envelope ("Gabor patch") or a random dot pattern (RDP) and will be asked to report whether a stimulus is moving to the left or to the right. Matching Tadin et al. [22], the grating will have a spatial frequency of 1 cycle per degree of visual angle. The radius of each grating will be defined as twice the standard deviation of the Gaussian envelope ($2\sigma$). We will use $\sigma$ values of 0.4˚, 1.0˚, 1.6˚, 2.2˚, and 2.8˚, resulting in grating of radii of 0.8˚, 2.0˚, 3.2˚, 4.4˚, and 5.6˚. The phase of the grating at stimulus onset will vary randomly from trial to trial and the gratings will move with a speed of 2.5˚/s to the left or the right. Stimuli are presented in the center of the screen against a gray background. Three different contrast levels will be presented by changing the gratings' transparency. Michelson contrasts, calculated as $\frac{I_{max}-I_{min}}{I_{max}+I_{min}}$ (where $I_{max}$ is the highest luminance (i.e., the brightest white) and $I_{min}$ is the lowest luminance (i.e., the darkest black) of the grating) for what we call the "high", "intermediate", and "low" contrast condition will be 99%, 8%, and 3%. Throughout the experiment $I_{max}$ and $I_{min}$ will always be equidistant from the background, i.e., all stimuli will have the same mean luminance as the background. We choose a relatively low contrast value (8%) for "intermediate" because Tadin et al. [22] showed the effect of size change to lie between 2.8% and 11%, with little qualitative difference between higher contrast levels. Our random dot pattern (RDP) stimulus consists of dots that are randomly placed in a circular aperture. The radius of the aperture will be 1˚, 2.5˚, 4˚, 5.5˚, or 7˚ (which, perceptually, resembles the size of our gratings) and the dot density will be 4 dots/deg$^2$ (i.e., between 12 and 615 dots). All dots will move coherently to the right or to the left with a constant speed of 2.5˚/s and dots that leave the aperture on one side will be reenter on the other side of the aperture. Dots have a diameter of 0.2˚. Half of the dots will be white and black each, to align with the grating stimulus' luminance variation. The same three different contrast levels ($I_{max}$ and $I_{min}$ now being the luminance of the white and the black dots) as for the gratings will be used. It should be noted that an exact size of the gratings cannot be determined, as their contrast fades out with increasing distance from the stimulus center and perceived size depends on the standard deviation and peak-contrast of the grating [33]. However, the two stimuli as we have specified them here appear similar in size upon visual inspection. Subjects will be asked to foveate a fixation dot at the center of the screen and to start each trial with a button press. The fixation dot will then disappear and the stimulus will be presented for a brief duration that will be adjusted for each subject individually based on a training session (see below for further information). Subsequently, the stimulus will either be masked for 220 ms with a random white noise stimulus (same mean luminance as the

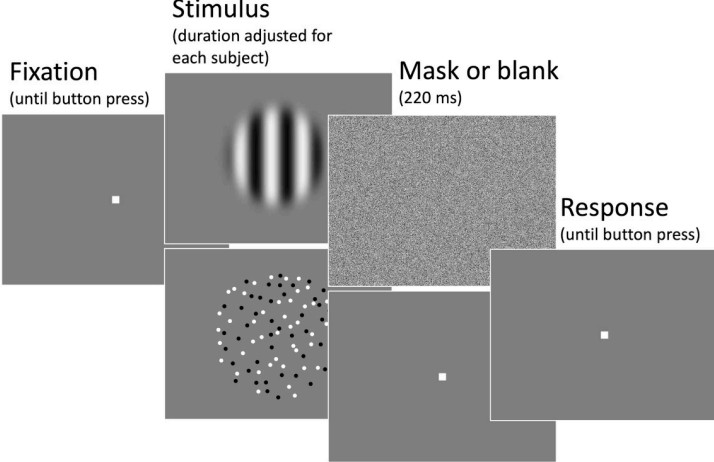

**Fig 1. Example trial sequence.** Subjects initiated each trial with a button press. Either a drifting grating or a random dot pattern moving to the left or to the right will then be briefly presented for a duration that will be adjusted for each subject individually to ensure a significant but not perfect performance of about 75% for the intermediate stimulus size. After the stimulus, either a white noise mask or a blank screen will be shown for 220 ms. Note that these are cartoons of the actual stimuli and that this figure has been optimized (e.g., in terms the grating's Gaussian envelope or the RDP's dot-size and number of dots) to illustrate the sequence of events.

background) or the screen will remain blank for the same duration. Participants will then report their perceived direction (left vs. right) by pressing the corresponding trigger button on the gamepad (see Fig 1). Trials on which the gaze deviated by more than 1.5˚ from the fixation dot position will be aborted and repeated at a later point during that block.

In total, there will be 60 different possible stimulus configurations: 2 stimulus types (grating and RDP) with 5 sizes and 3 contrast levels each, and either a white noise mask or a blank screen after the stimulus presentation. Each of these 60 configurations will be presented 48 times (2880 trials in total) in 12 blocks of 240 trials each. Within each block, every stimulus configuration will be presented 4 times in random order. We plan to collect the data from each subject in two separate session. To familiarize subjects with the task, they will undergo a training session prior to the first test session. Each subject will be presented with 10 blocks of 100 trials each (1000 trials in total), half of which will be with gratings and half with RDPs of varying sizes and contrast levels (though not necessarily the exact values that will be used during the experiment). The presentation duration of the stimulus will be varied during the training session, starting with long durations until subjects have understood the task well. Towards the end of the training session, one stimulus duration (<200 ms) will be determined manually for which the subject is able to perform the task well for all combinations of stimulus type, contrast levels, and sizes. Our experience from the pilot study has been that after performing around 600 trials, subjects are sufficiently familiarized with the task and their performance stabilizes. Typically, there are only one or two stimulus durations for each subject at which they perform reliably above chance level without showing ceiling effects in their performance. These duration values can easily be determined manually, because subjects' performance quickly approaches chance level or 100% correct responses if the duration is decreased or increased by just one or two frames (8.3 or 16.6 ms). Should no such stimulus configuration be found the participant will not enter the study and an alternate participant will be recruited.

## Data analysis

For each stimulus type, contrast level, size, and mask/no-mask condition we will calculate the percentage of correct responses (out of 48 presentations) for each subject as a measure of how well they were able to discriminate the stimulus direction. To investigate whether performance is affected by stimulus size, how this size-effect is influenced by contrast and post-stimulus masking, and whether these effects and interactions differ between stimulus types, we aim to determine the effect of each of the four independent variables on performance. We will do this by fitting a generalized linear mixed-effects model (GLMM) with STIMULUS TYPE, STIMULUS SIZE, CONTRAST, and MASK as fixed effects and SUBJECT as a random effect. The model will be fitted with binomial error structure and logit link-functions and therefore assess the probability of a correct response. Per participant we shall collapse the response across trials but separately for each combination of STIMULUS TYPE, STIMULUS SIZE, CONTRAST, and MASK, and hence the response will be a two-columns matrix with the number of correct and incorrect responses [34]. The model will include all interactions between STIMULUS TYPE, STIMULUS SIZE, CONTRAST, and MASK, up to fourth order as fixed effects. As random slopes it will include STIMULUS TYPE, STIMULUS SIZE, CONTRAST, and MASK and all their interactions up to third order, and also parameters for the correlations among the random intercept and slopes.

## Discussion

Drifting sinusoidal gratings and random dot patterns are frequently used and powerful stimuli to investigate motion perception in psychophysics [10, 22, 35–40], functional imaging [41–43], and electrophysiology [2, 44–47]. The popularity of these stimuli is due to the fact that they can be carefully created to isolate features of interest (e.g., direction of motion), with little confounding by other features (e.g., color) (see Rust & Movshon [48], for a discussion of the advantages of artificial, synthetic stimuli). However, studies rarely motivate their use of stimulus type and much less research has been dedicated to investigating whether there are relevant differences in how these two stimulus types are processed and perceived. It is a broadly accepted idea that early motion-sensitive areas of the primate cortex (e.g., V1) are predominantly concerned with local motion information, whereas later areas (e.g., MT) respond preferentially to global motion information [11, 36, 41]. This increase in spatial integration can be explained, at least partly, by the increase in receptive field size along the processing pathway from V1 to MT (e.g., [49]). Gratings require little spatial integration as all the available motion and orientation information can be extracted from any small portion of the stimulus. For RDPs, on the other hand, a small part of the stimulus might not contain any dots (depending on the dot density) or have a low signal-to-noise ratio (depending on coherence) so that spatial integration is necessary to perceive a well-defined motion direction. Additionally, RDPs do not suffer from the aperture problem [11] as the motion direction of dots is unambiguous. Thus, a plausible (but presumably oversimplistic) hypothesis would be that perception of gratings relies primarily on activity in V1 whereas that of RDPs requires activity in area MT. As Neri & Levi [50] have pointed out, different stages along the visual hierarchy might impose different constraints upon further processing which sometimes can be observed in the final behavioral output. It is in that sense that we suggest V1 and MT to have different roles in the processing of RDP and grating motion. The idea of different stages of the visual processing hierarchy imposing different constraints is supported by a study that varied contrast and the overall area of RDPs. While the dependence of motion sensitivity on stimulus area was well explained by linear integration across the entire stimulus, the dependence on contrast sensitivity was better explained by local motion detectors with contrast thresholds [37]. Based on

these findings, the authors suggest a two-stage model of motion processing: an initial stage consisting of contrast-sensitive local motion detectors (presumably V1) and a second stage that integrates across the detectors and is tuned for more complex motion patterns (presumably MT/MST). In summary our study will thus provide an important perceptual test of the hypothesis of a stimulus-dependent neural processing of visual motion information and guide further behavioral and neurophysiological research into how the building blocks of cortical information processing contribute to visual perception.

## Supporting information

**S1 Appendix. Description and results of pilot experiment.**
(PDF)

**S1 Dataset. Data from pilot experiment.**
(PDF)

## Acknowledgments

The authors thank Roger Mundry for statistical consultation, Cliodhna Quigley for helpful discussion, and Ann-Kristin Kenkel for help during data collection for the pilot study.

## Author Contributions

**Conceptualization:** Benedict Wild, Stefan Treue.

**Data curation:** Benedict Wild.

**Funding acquisition:** Stefan Treue.

**Investigation:** Benedict Wild.

**Methodology:** Benedict Wild, Stefan Treue.

**Project administration:** Stefan Treue.

**Resources:** Stefan Treue.

**Software:** Benedict Wild, Stefan Treue.

**Supervision:** Stefan Treue.

**Writing – original draft:** Benedict Wild.

**Writing – review & editing:** Benedict Wild, Stefan Treue.

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
