## [Decision Letter · Decision Letter 0]

25 Mar 2021

PONE-D-21-00723

Comparing the influence of stimulus size and contrast on the perception of moving gratings and random dot patterns

PLOS ONE

Dear Benedict Wild,

I am very sorry for the long delay in getting back to you. I had send the manuscript to the two previous reviewers but I only managed to get one review as the other reviewer dropped out in the process. Rather than delaying further I decided to go ahead based on the one review I obtained. The reviewer recommended major revision.

We look forward to receiving your revised manuscript.

Kind regards,

Markus Lappe

Academic Editor

PLOS ONE

Journal Requirements:

Reviewers' comments:

Reviewer's Responses to Questions

**Comments to the Author**

1. Does the manuscript provide a valid rationale for the proposed study, with clearly identified and justified research questions?

Reviewer #1: Partly

2. Is the protocol technically sound and planned in a manner that will lead to a meaningful outcome and allow testing the stated hypotheses?

Reviewer #1: Partly

3. Is the methodology feasible and described in sufficient detail to allow the work to be replicable?

Reviewer #1: No

4. Have the authors described where all data underlying the findings will be made available when the study is complete?

Reviewer #1: No

5. Is the manuscript presented in an intelligible fashion and written in standard English?

Reviewer #1: Yes

6. Review Comments to the Author

You may also provide optional suggestions and comments to authors that they might find helpful in planning their study.

Reviewer #1: The study protocol comparing the influence of stimulus size and contrast on the perception of moving gratings and random dot patterns by Wild and Treue tackles the interesting question to what extent different stimulus parameters (size, contrast) affect direction of motion discrimination and what this can tell us potentially about motion processing in the brain. The question is much studied but worthwhile, especially given the direct comparison of grating and dot stimuli and study of mask effects.

My main concerns are that (1) the framing of the research hypothesis remains imprecise to the reader, (2) some methods/stimulus parameters are missing for replication, (3) the statistics need a little more thought and (4) no info is given about _where_ the data will be available upon completion.

(1) Hypothesis and discussion of previous literature:

It is not entirely clear what the specific hypothesis is that the authors want to test with their study and how their expected results will relate to this.

In the abstract, the authors start with the view that different stages in the motion processing pipeline - V1 and MT - make specific, independent contributions to motion perception. Then say that it more depends on the stimulus type and perceptual history. But they and others use gratings and dots precisely to tap into V1 and MT processing respectively?

In the introduction, the authors then go on to say that the contribution of each area is "highly flexible", which might mean interchangeable. But in the same sentence, the authors also say the contribution of an area to motion perception again depends on particular stimulus features without specifying those features. I am sorry if I missed something but this seems to me contradictory.

The authors should set out the view of the motion processing pipeline they seek to test more clearly (serial and different read-outs at different cortical levels or parallel pathways through these areas) and underpin this with the literature they cite.

The study protocol also needs a clearer discussion to explicitly name what outcomes are expected if the hypothesis is true and what the alternative framework might be if not.

Although the exploration of the mask effect is interesting, I am also missing an explanation about what the mask effect is on the motion processing pipeline.

(2) Missing stimuli, display parameters /methods:

- mean luminance of the screen under experimental conditions for the different stimulus conditions and during different phases of a trial (esp. related to the mask). It is important that this does not change.

- where is the stimulus placed, centrally like in the figure?

- is there a head mount?

- training phase - how is the end (the stimulus duration) determined? Is there a systematic approach / a clear end point.

- Participants:

(i) was there a power analysis done on the pilot data to determine the exactly 16 participants? Given there is pilot data which allows estimation of the effect size, a power analysis would be appropriate.

(ii) Why 18-35 years only rather than adults?

(iii) why are particularly runny noses or some coughing exclusion criteria? If s this about COVID hygiene rules or generally healthy, please state.

(3) Analysis: given that the literature talks about an optimal stimulus size either side of which performance drops, doing a Pearson's correlation that assumes a systematically falling or rising relationship is a bit odd as the first analysis of the data. The rmANOVA seems more appropriate as first analysis to see if there are any differences.

This requires a bit more thought, justification and perhaps some illustration from the pilot data.

(4) I could not find a statement _where_ the to-be-collected data will be made available.

Minor points:

(5) page 3, line 25 (Introduction). Please name these "key roles".

(6) page 3, line 46: "The authors concluded"

(7) p.8. line 175, in an analysis section, "discriminate" for this task would be more appropriate than "perceive".

(8) p.9. Discussion, first paragraph: "little research has been dedicated to investigating whether there are relevant differences in how these two stimulus types are processed and perceived". I think this statement does not adequately reflect the cited literature well nor the wider range of psychophysics and neurophysiological studies on motion processing with grating, bar and RDP stimuli.

(9) p.9. line 220: "dependence of contrast sensitivity". I think the authors mean "on" here, which does give the sentence a different meaning.

7. PLOS authors have the option to publish the peer review history of their article (what does this mean?). If published, this will include your full peer review and any attached files.

Reviewer #1: No

---

## [Author Response · Author response to Decision Letter 0]

21 May 2021

We have also uploaded a pdf with formatted responses, which might be easier to read.

Reviewer #1: The study protocol comparing the influence of stimulus size and contrast on the perception of moving gratings and random dot patterns by Wild and Treue tackles the interesting question to what extent different stimulus parameters (size, contrast) affect direction of motion discrimination and what this can tell us potentially about motion processing in the brain. The question is much studied but worthwhile, especially given the direct comparison of grating and dot stimuli and study of mask effects.

My main concerns are that (1) the framing of the research hypothesis remains imprecise to the reader, (2) some methods/stimulus parameters are missing for replication, (3) the statistics need a little more thought and (4) no info is given about _where_ the data will be available upon completion.

(1) Hypothesis and discussion of previous literature:

It is not entirely clear what the specific hypothesis is that the authors want to test with their study and how their expected results will relate to this.

In the abstract, the authors start with the view that different stages in the motion processing pipeline - V1 and MT - make specific, independent contributions to motion perception. Then say that it more depends on the stimulus type and perceptual history. But they and others use gratings and dots precisely to tap into V1 and MT processing respectively?

In the introduction, the authors then go on to say that the contribution of each area is "highly flexible", which might mean interchangeable. But in the same sentence, the authors also say the contribution of an area to motion perception again depends on particular stimulus features without specifying those features. I am sorry if I missed something but this seems to me contradictory.

The authors should set out the view of the motion processing pipeline they seek to test more clearly (serial and different read-outs at different cortical levels or parallel pathways through these areas) and underpin this with the literature they cite.

The study protocol also needs a clearer discussion to explicitly name what outcomes are expected if the hypothesis is true and what the alternative framework might be if not.

We have rewritten large sections of the introduction to specify our hypotheses in more detail [specifically in line 96ff].

Although the exploration of the mask effect is interesting, I am also missing an explanation about what the mask effect is on the motion processing pipeline.

We have explained our motivation for including this variable in more detail [lines 100ff].

(2) Missing stimuli, display parameters /methods:

- mean luminance of the screen under experimental conditions for the different stimulus conditions and during different phases of a trial (esp. related to the mask). It is important that this does not change.

We have added a statement “Throughout the experiment I_max and I_min will always be equidistant from the background, i.e., all stimuli will have the same mean luminance as the background.“ [lines 182-184].

- where is the stimulus placed, centrally like in the figure?

Yes. We have added the information in the Methods section [line 178].

- is there a head mount?

Yes. We have added the information in the Methods section [line 165].

- training phase - how is the end (the stimulus duration) determined? Is there a systematic approach / a clear end point.

We have added more information about our training procedure in the Methods section [lines 228 ff].

- Participants:

(i) was there a power analysis done on the pilot data to determine the exactly 16 participants? Given there is pilot data which allows estimation of the effect size, a power analysis would be appropriate.

We ran a power analysis and have described our decision for the number of participants to be recruited in the Methods section [lines 117-130].

(ii) Why 18-35 years only rather than adults?

We decided to focus on young adults because research by Betts et al (2005, Neuron) has shown that performance in this type of task varies with age. We have also added this explanation in the manuscript [line 136 ff]

(iii) why are particularly runny noses or some coughing exclusion criteria? If s this about COVID hygiene rules or generally healthy, please state.

Yes, this statement refers to additional precautions we are currently taking. We have added this information to the manuscript [line 133].

(3) Analysis: given that the literature talks about an optimal stimulus size either side of which performance drops, doing a Pearson's correlation that assumes a systematically falling or rising relationship is a bit odd as the first analysis of the data. The rmANOVA seems more appropriate as first analysis to see if there are any differences.

This requires a bit more thought, justification and perhaps some illustration from the pilot data.

Upon consultation with a statistician, we have decided to analyze the data with a generalized linear mixed-effects model and have adjusted the Data Analysis section in the manuscript accordingly [lines 248 ff].

(4) I could not find a statement _where_ the to-be-collected data will be made available.

We will make the data available as a repository on our lab’s account on the German Neuroinformatics (G-Node) server). We have added a statement about this at the end of the methods section. [line 271 ff.]

Minor points:

(5) page 3, line 25 (Introduction). Please name these "key roles".

We have changed the sentence as follows: “The often implicitly assumed “standard model” of motion processing assigns key roles in representing direction and speed to neural activity in primary visual cortex (V1) and the middle temporal (MT) area.” [lines 25-27].

(6) page 3, line 46: "The authors concluded"

We have corrected the typo.

(7) p.8. line 175, in an analysis section, "discriminate" for this task would be more appropriate than "perceive".

We have replaced “perceive” with “discriminate” [line 248].

(8) p.9. Discussion, first paragraph: "little research has been dedicated to investigating whether there are relevant differences in how these two stimulus types are processed and perceived". I think this statement does not adequately reflect the cited literature well nor the wider range of psychophysics and neurophysiological studies on motion processing with grating, bar and RDP stimuli.

We have mitigated the statement by changing it to: “However, studies rarely motivate their use of stimulus type and much less research has been dedicated to investigating whether there are relevant differences in how these two stimulus types are processed and perceived.” [line 308].

(9) p.9. line 220: "dependence of contrast sensitivity". I think the authors mean "on" here, which does give the sentence a different meaning.

We have corrected the typo.

We thank the reviewer for her or his comments.

---

## [Editor Report · Decision Letter 1]

28 May 2021

Comparing the influence of stimulus size and contrast on the perception of moving gratings and random dot patterns

PONE-D-21-00723R1

Dear Dr. Wild,

We’re pleased to inform you that your manuscript has been judged scientifically suitable for publication and will be formally accepted for publication once it meets all outstanding technical requirements.

Kind regards,

Markus Lappe

Academic Editor

PLOS ONE
---

## [Editor Report · Acceptance letter]

10 Jun 2021

PONE-D-21-00723R1 

Comparing the influence of stimulus size and contrast on the perception of moving gratings and random dot patterns - a registered report protocol 

Dear Dr. Wild:

I'm pleased to inform you that your manuscript has been deemed suitable for publication in PLOS ONE. Congratulations! Your manuscript is now with our production department. 

Kind regards, 

on behalf of

Dr. Markus Lappe 

Academic Editor

PLOS ONE